# Intra Articular Injection of Autologous Microfat and Platelets-Rich Plasma in the Treatment of Wrist Osteoarthritis: A Pilot Study

**DOI:** 10.3390/jcm11195786

**Published:** 2022-09-29

**Authors:** Alice Mayoly, Marie Witters, Elisabeth Jouve, Cécilia Bec, Aurélie Iniesta, Najib Kachouh, Julie Veran, Fanny Grimaud, Anouck Coulange Zavarro, Rémi Fernandez, David Bendahan, Laurent Giraudo, Chloé Dumoulin, Christophe Chagnaud, Dominique Casanova, Florence Sabatier, Régis Legré, Charlotte Jaloux, Jérémy Magalon

**Affiliations:** 1Department of Hand and Limb Reconstructive Surgery, Hôpital de la Timone, Assistance Publique-Hôpitaux de Marseille, 13005 Marseille, France; 2Pharmacometry, Clinical Investigation Center—Center for Clinical Pharmacology and Therapeutic Evaluations (CIC-CPCET), Clinical Pharmacology and Pharmacovigilance Department, Hôpital de la Timone, AP-HM, 13005 Marseille, France; 3Therapy Cell Laboratory, Hôpital de la Conception, AP-HM, INSERM CIC BT 1409, 13005 Marseille, France; 4Radiology Department, Hôpital de la Conception, Assistance Publique-Hôpitaux de Marseille, 13005 Marseille, France; 5Biological and Medical Magnetic Resonance Center, 13005 Marseille, France; 6Department of Plastic and Reconstructive Surgery, Hôpital de la Conception, Assistance Publique-Hôpitaux de Marseille, 13005 Marseille, France; 7C2VN, INSERM 1263, INRA 1260, Aix-Marseille University, 13005 Marseille, France

**Keywords:** PRP, wrist osteoathrisis, microfat, platelet-rich plasma, biotherapy

## Abstract

No injection treatment has been proven to be effective in wrist osteoarthritis. When conservative measures fail, its management involves invasive surgery. Emergence of biotherapies based on adipose derived stem cells (ADSC) offers promising treatments for chondral degenerative diseases. Microfat (MF) and platelets-rich plasma (PRP) mixture, rich in growth factors and ADSC could be a minimally invasive injectable option in the treatment of wrist osteoarthritis. The aim of this uncontrolled prospective study was to evaluate the safety of a 4 mL autologous MF-PRP intra-articular injection, performed under local anesthesia. The secondary purpose was to describe the clinical and MRI results at 12 months of follow-up. Patients’ data collected were: occurrence of adverse effects, Visual analog scale (VAS), Disabilities of the Arm, Shoulder and Hand score (DASH) and Patient-Rated Wrist Evaluation (PRWE) scores, wrist strength, wrist range of motion and 5-level satisfaction scale. No serious adverse event was recorded. A statistically significant decrease in pain, DASH, PRWE and force was observed at each follow-up. Our preliminary results suggest that intra-articular autologous MF and PRP injection may be a new therapeutic strategy for wrist osteoarthritis resistant to medical symptomatic treatment prior to surgical interventions.

## 1. Introduction

Wrist osteoarthritis (OA) is a progressive non-inflammatory joint disease involving radiocarpal and/or midcarpal cartilages. It is one of the most common afflictions encountered by hand surgeons. Although it can be well tolerated for years, it leads to long-term severe functional impairments including pain and loss of strength and motion. Wrist OA is usually secondary to post-traumatic sequelae with two main etiologies: malunions of the distal radial articular surface due to intra-articular fractures, and disturbance of carpus biomechanics due to peri-scaphoid lesions (scapho-lunate ligament rupture named scapho-lunate advanced collapse or SLAC, and scaphoid non-union named scaphoid non-union advanced collapse or SNAC). Rarely, wrist OA is due to idiopathic causes like carpal avascular necrosis (Kienböck’s or Preiser’s disease) or congenital wrist abnormalities (Madelung’s deformity) [1,2,3,4]. Conservative treatments by splints, analgesics and anti-inflammatory drugs are considered the first line of treatment whereas surgical procedures are indicated in cases of refractory pain [3,5]. Corticosteroid and hyaluronic acid injections, which are widely used in gonarthrosis, have limited evidence of their effectiveness in wrist OA [6,7,8]. Surgical treatments can reduce patients pain and preserve wrist function but results are not always optimal [9,10,11,12,13]. 

There is a real therapeutic gap between conservative treatments and invasive surgeries in the treatment of wrist OA. Finding a minimally invasive therapy that would delay the need for surgical treatment is a challenge in the management of this condition. Biological injectable options and cell therapy have recently emerged in treatment of chondral degenerative diseases with promising results. Among them, Platelet-Rich-Plasma (PRP), defined as an autologous plasma suspension of platelets, is used in reparative and regenerative processes due to the high concentration of growth factors contained in platelets [14,15,16]. PRP has been the subject of increasing clinical interest in the treatment of chondral degenerative diseases with satisfactory results particularly in gonarthrosis [16,17,18,19,20]. Another innovative option is related to the use of adipose tissue which is easily harvested with liposuction. Stromal vascular fraction (SVF) cells contained in adipose tissue include mesenchymal multipotent stem cells (Adipose-derived stem cells: ADSCs) which are capable of differentiating into cartilage cells [21,22,23,24,25]. The term microfat (MF) was recently described and relates to a dedicated adipose tissue harvesting procedure using a multiperforated cannula with holes of 1 mm selecting only small fat lobules [26]. MF has better trophic and mechanical properties than conventionally harvested adipose tissue. It allows better adhesion and migration of multipotent stem cells from adipose tissue and has better fluidity which allows it to be injected intra-articularly with small gauge needles [27].

The combination of autologous microfat and PRP (MF-PRP), respectively rich in multipotent stem cells and growth factors, aims to create an optimal environment for cartilage cell regeneration. This regenerative product has been previously described in racehorses with good results associated with efficacy as assessed by significant reduction in lameness and an early return to competition [28]. Given that this product requires a minimally invasive procedure performed under local anesthesia in a half-outpatient setting, the combination of autologous MF-PRP is a promising therapy for wrist OA. As far as the regulatory aspect is concerned, certain issues related to the classification of this product as an Advanced Therapy Medicinal Product (ATMP) have been circumvented because of the exhaustive biological characterisation described previously [29].

Our hypothesis is that standardized intra-articular injection of this innovative treatment (Autologous MF-PRP) could relieve pain and improve function in patients with wrist OA resistive to previous well-conducted conservative treatment. The aim of this study was to assess feasibility and safety of autologous MF-PRP intra-articular injection in the treatment of wrist OA in these patients. The secondary purpose was to provide preliminary clinical results and Magnetic Resonance Imaging (MRI) findings after a 12 months follow-up.

## 2. Materials and Methods

### 2.1. Study Design and Patient’s Selection

A single site uncontrolled feasibility trial of phase I–IIa ran from June 2017 to February 2019 and was performed in collaboration between the Hand and Limb Reconstructive Surgery department of La Timone University Hospital, the Plastic and Reconstructive Surgery Department of La Conception University Hospital, and the Cell Therapy Department of La Conception University Hospital. After a first screening, the suitability of patients for inclusion was assessed by a hand surgeon, according to the inclusion and exclusion criteria presented in Table 1. The protocol was approved by a national ethics committee (authorization #16-65 from Comité de Protection des Personnes Sud Méditerranée #1) and national health regulatory authority (Agence Nationale de Sécurité du Médicament et des Produits de Santé, authorization #160879A-12) and registered in Clinicaltrials.gov website (NCT03164122 and EudraCT # 2016-002648-18). The study was carried out according to the Declaration of Helsinki and with the principles of Good Clinical Practice. All patients gave written informed consent before their participation.

### 2.2. Surgical Procedure

The procedure consisted of two consecutive surgical steps performed under local anesthesia in an operation theatre and the preparation of the experimental products performed in the cell therapy department. The first surgical step consisted of MF and blood harvesting and the second step correspond to the reinjection of the autologous MF-PRP product inside the wrist. The entire procedure was carried out over half a day in an outpatient setting.

### 2.3. Blood and Adipose Tissue Harvesting

The harvesting procedures were performed after skin decontamination (antiseptic foaming solution, rinsing with sterile water, drying, antiseptic foaming solution, rinsing with sterile water, drying and alcoholic dermal antiseptic). 18 mL of blood was collected by venipuncture using a 21-gauge needle filling one 20-mL syringe containing 2 mL of Anticoagulant Citrate Dextrose Solution, Solution A (ACD-A) (Fidia, Abano Terme, Italy). The harvesting of adipose tissue was performed on the inner side of the knees after infiltration with a local anesthetic solution composed of 140 mL of injectable serum and 60 mL of xylocaine^®^ (10 mg/mL) with adrenaline (0.005 mg/mL). 50 mg of adipose tissue were harvested using a Hapifat^®^ kit (Benew Medical, Melesse, France). A St’rim cannula (14 gauge, 2-mm external diameter, 8 holes of 0.58 mm square) was both connected to a 10 mL syringe and a purification Puregraft 50^®^ bag through a Fat Lock System^®^ (Figure 1). Both products were immediately packed in a sterile bag before being transferred to the GMP cell therapy facility where all procedures were performed within a class A microbiological safety cabinet.

### 2.4. Preparation of Experimental Products: MF-PRP

The blood was transferred into the Hy-tissue 20 PRP device (Fidia, Abano Terme, Italy) before centrifugation using the Omnigrafter 3.0 (Fidia, Abano Terme, Italy) and PRP Large Volume Cycle (3200 rpm for 10 min). All plasma was recovered using a 10 mL syringe through the Push-out system. 300 µL of whole blood and PRP preparation were sampled to determine platelet, leukocyte and RBC concentrations.

The harvested adipose tissue was purified twice using a 1:1 rinsing with saline solution [30] allowing elimination of fluid excess, lipid, blood cells, and fragments through filtration by the Puregraft bag membrane.

Final experimental products were packaged in two 5 mL syringes, each containing 2 mL of PRP and purified MF. These 2 syringes were connected to each other by a 3-way valve in closed position.

### 2.5. Biological Characterization of Injected Products

Cell count: whole blood and each PRP preparation were sampled to determine platelets, leukocytes and red blood cells counts using automated hematology blood cell analyzers Sysmex XN-10 (Sysmex, Kobe, Japan) in accordance with recently published guidelines [31].

Microbiological assay: 250 µL of PRP or MF were sampled in Bactec culture bottles (Peds Plus Aerobic/F and Plus Anaerobic/F culture vials, containing each 40 mL of medium). The Bactec method (Becton Dickinson, Sparks, MD, USA) uses a computer-controlled incubation/detection system. The media used contained proprietary factors designed to inactivate a wide variety of antibacterial and antifungal agents. Bactec culture bottles were incubated at 37 °C for a total of 10 days, and automated readings were taken every 10 min. Detection of organisms resulted in an audible alarm and automatic recording at detection.

Mixed products Growth Factors Release Measurement. 500 μL of MF and PRP were mixed and then placed in a 48-well collagen coated plate (CellAffix, APSciences, Columbia, MD, USA) and incubated for 30 min at 37 °C with 5% CO_2_ to allow contact with collagen. After this incubation, 500 μL of non-supplemented Dulbecco’s Modified Eagle Medium (DMEM) (Gibco, Thermo Fisher Scientific, Waltham, MA, USA) was added to the mixture and incubated for 24 h at 37 °C with 5% CO_2_. After 24 h the samples were centrifuged (Multifuge Heraus 3 S-R centrifuge, Thermo Scientific, Indianapolis, IN, USA) at 1500 rpm for 5 min to remove MF, and samples were stored at −80 °C until used for analysis. A combination of 12 cytokines and growth factors classified as inflammatory (Interleukin-1β, Tumor Necrosis Factor-α, Interleukin-6, Inteferon γ), anti-inflammatory (Interleukin-1 Receptor antagonist, Interleukin-10) and regenerative (Platelet Derived Growth Factor AA-BB or AB-BB, Vascular Endothelial Growth Factor, Nerve Growth Factor, Epidermal Growth Factor, Fibroblast Growth Factor 2, Transforming Growth Factor β1) were measured using a Magpix instrument (Luminex xMAP Technology, Luminex Inc., Austin, TX, USA) allowing simultaneous measurement of the different analytes in small sample volume.

### 2.6. Intra-Articular Injection of MF-PRP

The two 5 mL syringes were connected and mixed by gently shaking back and forth ten times to obtain a final homogeneously mixed product of 4 mL. After 4 steps of skin decontamination and local anesthesia with xylocaine^®^ (10 mg/mL) without adrenaline at the dorsal side of the wrist, 4 mL of MF-PRP was injected with a 16-gauge needle into the radiocarpal joint near the scapholunate ligament. The injection site was located 10 mm distal and 5 mm ulnar to the radial tubercle. The exact location of the needle was verified under fluoroscopic guidance (Zhiem Imaging solo E5830-SD-H6 Ziehm Imaging GmbH, Nuremberg, Germany). Pain according to the Visual Analog Scale (VAS) was also recorded during intra-articular injection. An analgesic immobilisation procedure with a specific cast providing compression, stabilization and cryotherapy (IGLOO^®^, IGPO, Implants Service Orthopédie, 91130 Ris-Orangis) was immediately put in place for 7 days. Patients went home with instructions to limit the overuse of the wrist and to use paracetamol or ice on the injected area to relieve pain if necessary. Non-steroidal anti-inflammatory drugs were prohibited for 7 days following the injection. During the follow-up, no treatment restriction was applied and, thereafter, a gradual return to normal sporting or recreational activities was tolerated.

#### 2.6.1. Evaluation Tools and Follow-Up

Patients were prospectively assessed at baseline and at 7 days and 1, 3, 6 and 12 months after injection. The primary endpoint was the safety of the treatment evaluated by the occurrence of adverse events up to one month after injection. Secondary clinical endpoints were assessed at 3, 6 and 12 months and included subjective pain rating by a VAS (0–100 mm), functional evaluation utilising the Disabilities of the Arm, Shoulder and Hand score (DASH) and the Patient-Rated Wrist Evaluation (PRWE), objective wrist strength measured by the Jamar hydraulic Hand dynamometer (average score of 3 consecutive measurements), objective wrist range of motion measured using a goniometer and patient’s satisfaction rated on a 5-level scale: Very dissatisfied, Dissatisfied, Neutral, Satisfied and Very satisfied.

#### 2.6.2. MRI Acquisition and Analysis

MRI examination was performed on a Verio 3 T Siemens MRI scanner (Siemens, Munich, Germany) with an 8-channel transmitter/receiver flexible coil wrapped around the wrist. Patients were in the supine position with the hand attached to a plastic support located a few centimetres above the abdominal wall. The carpal bone region was scanned coronally using a 2D T1 TSE sequence (TR = 500 ms, TE = 11 ms, voxel size = 0.4 × 0.4 × 2.5 mm^3^), a 2D fat saturated TSE sequence (TR = 3000 ms, TE = 46 ms, voxel size = 0.3 × 0.3 × 2.5 mm^3^) and a 3D VIBE sequence (TR = 10 ms, TE = 4 ms, voxel size = 0.5 × 0.5 × 0.5 mm^3^).

Pre- and post-injection (12 months) MRI were anonymized and randomized. The 3D sequences after reformatting in the same section were read in a blinded way by a senior radiologist specialized in osteoarticular imaging. Two parameters were studied between pre and post MRIs: the structural modification of the cartilage of the joint by analysing change in the thickness of the radioscaphoid space in the presence of oedema. The radioscaphoid space was measured between the line joining the subchondral bone plate of the scaphoid to the radius (hyposignal line under the cartilage of the 2 bones) facing sector II and III of the scaphoid [32]. Both parameters were classified as improved, impaired or stable for each patient.

The degree of confidence of the radiologist was scored between 0 and 2 using a Likert scale (0 = uncertain, 1=possible and 2 = certain).

#### 2.6.3. Statistical Analysis

The sample size was fixed at 12 patients without formal calculation (no available data) based on typical sample sizes in early evaluation of a new biological and cell therapy.

Quantitative variables are described as mean ± standard deviation (SD) with median [minimum–maximum] in Appendix A. Categorical variables are described as numbers and percentages. Absolute changes at 3-, 6- and 12-months post-treatment are calculated from baseline. Safety analyses were only descriptive. To assess preliminary efficacy of this treatment, changes were tested (H0: change = 0) using the Wilcoxon signed rank test. A stepdown Bonferroni correction was applied to reduce the chance of a type one error for multiplicity. Responder patients were defined according to the minimal clinically important differences (MCID) of scores and a test of binomial proportion (*p* = 0.5) was performed. Longitudinal changes on the DASH of 10.83 points and on the PRWE of 11.5 points represent the MCID [33,34]. The proportion of responder patients on wrist strength was compared between affected vs normal wrist using the McNemar exact test. 

## 3. Results

### 3.1. Characteristics of Patients

Twelve patients (8 men and 4 women) aged of 53.8 ± 14.8 years were included in the study and followed until 12 months. They had wrist OA of grade 3 or 4 according to Kellgren and Lawrence classification with a mean duration of disease of 6.6 ± 9.1 years since diagnosis. The wrist OA was due to SLAC (n = 4), SNAC (n = 4) and radius malunions (n = 4). The radiocarpal joint only was affected in 8 cases, while both radial and mid-carpal joints were involved in 4 cases. The dominant hand was affected in 7 patients. The baseline characteristics of the patients are presented in Table 2.

### 3.2. Biological Characteristics of Mixed Products

Table 3 summarises biological characteristics of MF-PRP. The final average volume injected of the mixed product was 3.6 ± 0.4 mL. The injected experimental product contained 741 ± 162 millions of platelets representing 95.3 ± 2.8% of the injected PRP cells compared to red blood cells (4.5 ± 2.7%) and leukocytes (0.1 ± 0.1%) corresponding to a high purity PRP according to the DEPA (Dose of injected platelets, Efficiency of production, Purity of the PRP, Activation of the PRP) classification [35]. The main regenerative growth factor (GF) released by the MF-PRP mix were fibroblast growth factor 2 (FGF-2) (6828 ± 3345 pg/mL), transforming growth factor β1 (TGF-β1) (5019 ± 2891 pg/mL) while the IL-6 inflammatory cytokine was also highly released (4669 ± 3187 pg/mL). However, the presence of the anti-inflammatory (interleukin-1 receptor antagonist) IL1Ra cytokine (155 ± 149 pg/mL) was detected with a mean IL1Ra/IL1-β ratio of 88.7 ± 108.4. The main nonconformity observed was the absence of sterility on MF in 2/12 batches.

### 3.3. Safety

No serious adverse event was recorded. The main side-effect was pain at the harvesting adipose tissue sites with a mean VAS value of 28.3 ± 15.9 (0–50) seven days after injection. This pain was relieved by grade 1 analgesics and completely resolved after one month follow-up. Half of the patients reported no pain or mild pain (VAS less than 3) during intra articular injection. Two patients reported moderate pain (VAS from 3 to 6) and four patients reported pain greater than 6. In all cases, the pain disappeared completely within one minute after the injection and all patients were able to resume their regular activities one week after the injection. Bacteriological contamination of the MF was reported in two batches. The bacterial species found in both samples were Staphylococcus epidermidis in one and propionibacterium acnes in the other. No clinical symptoms were observed but both patients were treated with antibiotics for 10 days as a precaution.

### 3.4. Clinical Assessment

A statistically significant improvement in pain according VAS was observed at each follow-up (Figure 2 and Appendix A) with a mean of 58.3 ± 13.2 at baseline decreasing to 25.8 ± 15.2 at one year. Seven patients had an improvement in pain greater than 30 at one year after injection. DASH and PRWE scores were also significantly improved at each follow-up and the MCID was reached in 9 (75%) of patients for the DASH and 11 (91.7%) for the PRWE at 12 months. A statistically significant increase in strength of the injured wrist was observed at each follow-up with a mean change of 8.3 ± 6.9 kg at one year from baseline (Figure 3 and Appendix A) versus 5.4 ± 4.9 kg for the normal wrist. Comparison between patients with radiocarpal OA and both radio- and midcarpal OA did not reveal any significant difference on VAS pain, DASH and PRWE scores and wrist strength (Appendix A). The injured wrist treated with MF-PRP mix also significantly improved ulnar inclination up to 6 months with persistence at 12 months whereas the normal wrist did not change over time. No significant increase was observed for flexion, extension or radial inclination of the injured wrist. Regarding patient satisfaction at one year, 9 were satisfied or very satisfied and 3 patients were neutral.

### 3.5. MRI Assessment

Qualitative MRI analysis of cartilage structural changes at 12 months showed improvement in 3 patients (25%) with a Likert confidence scale of 1 (n = 2) and 0 (n = 1). No change was noted in 6 patients (50%) with Likert confidence scale of 2 (n = 1), 1 (n = 2) and 0 (n = 3) while possible impairment was observed in 3 patients (25%) with uncertainty for all patients. Regarding the variation in oedema, the radiologist confidence level was higher (11/12 assessment score 2 on the Likert scale): 3 (25%) patients were improved, 7 (58 %) showed no difference while 2 (17%) patients were impaired. Figure 4 illustrates the pre- and post-injection MRIs of one patient showing improvement in cartilage structural changes.

## 4. Discussion

The aim of this study was to evaluate the feasibility and the safety of intra articular injection MF-PRP in treatment of wrist OA. The only side effect reported in our cohort was pain at the harvesting adipose sites which was relieved with grade 1 analgesics and completely resolved after 1 month. This adverse event is commonly observed after any liposuction and also reported in other similar studies [36,37,38,39]. This is consistent with the good tolerance observed after injection of MF-PRP mixture into the joints of racehorses [28]. More generally, no infectious or neoplastic complications related to intra-articular injection of any orthobiological product (PRP, ADSCs, SVF or unfractionated adipose tissue) have been reported in the literature [40,41,42,43] whereas the most common complications reported were early pain and swelling of the injected joint [36,40,44,45,46,47,48]. The latter side effect is usually observed after joint injection, regardless of the product injected. In fact, in comparative studies evaluating SVF or ADSC injection versus placebo or hyaluronic acid, pain and swelling of the injected joint were reported in both groups [37,46,49,50,51].

Pain and swelling are common after injection in the knee but rather rare in carpo-metacarpal injection [36]. Delayed tenosynovitis and tendonitis at 6–8 weeks of injection have been described by one author in 20% of patient after injection of SVF and PRP in various joints and mainly concerned elderly patients [40].

A bacteriological contamination of MF was identified in two cases without clinical consequence. As a precaution, we treated these patients with antibiotics after medical advice from infectious disease specialists. Although this aspect raises concerns about patient safety, this type of contamination has been previously described by Louis et al. also using MF (4/30 products) and Garza et al. using SVF (2/26 products) [52,53]. None of these events were associated with clinical infections.

Many regenerative therapies, based on bone marrow or adipose tissue-derived multipotent stem cells capable of differentiating into cartilage cells have been evaluated for the treatment of OA. Adipose tissue is considered a valuable source of multipotent stem cells with up to 40 times the concentration of bone marrow [54]. The most popular strategy was to directly inject a homogeneous suspension of multipotent stem cells (MSCs). Several comparative studies have recorded the superiority of this treatment over placebo or hyaluronic acid in the treatment of knee OA [46,49,50,51]. This technique requires several complex and expensive purification and expansion steps by culture in an approved manufacturing center lasting two to three weeks [55]. In contrast, the preparation of MF can be done in one hour, allowing for liposuction and reinjection to be performed in the same procedure. The procedure requires the use of a dedicated cannula to select small (600 µm) fat lobules followed by a purification step using the Puregraft^®^ bag membrane required to remove oily substances and blood residues [26]. Although the manufacturing slightly differs with microfragmented fat (MFAT) [36,38,41,56,57,58,59], both techniques generally select small fat lobules that retain intact MSCs in the vascular niche of the adipose stroma during the preparation process [59]. 

Regarding the efficacy of autologous unfractionated adipose tissue for pain relief, several authors have reported injections in OA of the thumb carpometacarpal joint, but this has never been performed in wrist OA. In 2017, Herold et al. were the first to study autologous unfractionated adipose tissue obtained by liposuction and centrifugation in the treatment of carpometacarpal OA of the thumb [60]. The mean change at 12 months from baseline in pain at stress according in VAS measured pain with movement was 53, 20, and 29 mm for stage 2, 3 and 4 according to Eaton /Littler classification, respectively. Haas et al. also treated thumb carpometacarpal OA from stage 1 to 3 (stage 3 = 58%) using unfractionated adipose tissue without centrifugation step and reported an average reduction of pain under stress of 29 mm at 12 months [38]. More recently, Froschauer et al., reported long term effect of unfractionated adipose tissue without centrifugation in stage 2 and 3 thumb carpometacarpal with the main reduction of pain being 50 mm at two years after the injection [39]. All these results are consistent with the pain VAS improvement reported in our study (32.4 ± 17.8 mm 12 months after injection) where all treated patients had advanced wrist OA (grade 3 or 4 according to Kellgren and Lawrence).

None of the studies mentioned prior used combined adipose tissue with PRP. The use of MF-PRP should improve clinical outcomes as it has been reported that PRP increases MSC proliferation and chondrogenic differentiation in vitro [16,61,62,63,64,65] and may improve cartilage regeneration. These properties combined with the release of platelet growth factors participating in the tissue healing process and inducing an anti-inflammatory effect in chondral disease could be important in the treatment of OA [66,67,68]. From a clinical perspective, only two studies reported the use of PRP combined with unmanipulated adipose tissue in knee OA. Both Louis et al. and Pintat et al. reported similar clinical improvement in WOMAC scores of approximatively 20 points at 6 months which was maintained up to one year [52,69]. We observed a similar improvement in the functional status of the wrist up to one year with a mean change of 20 ± 11.8 and 35.7 ± 19.5 in DASH and PRWE scores respectively. Interestingly, Louis et al. compared the injection of MF mixed or not with PRP, and found no significant difference between the experimental groups. This raises the question of the necessity of using PRP in the study product, although Louis et al. conceded that their trial had a limited number of patients treated, and could have been underpowered. Both studies also reported MRI findings showing a decrease in relaxation time, a reduction in the severity of the grade of OA and an improvement in joint space, but without reaching statistical significance. Nevertheless, performing MRI of the wrist was not an easy task compared to MRI of the knee. The small size of the carpal bones requires higher accuracy and image quality to be able to analyze a structural change of the cartilage. In addition, due to the constant micromovements of the carpal bones, an image registration process was required to make the pre- and post-injection MRI slices stackable. This process deteriorated the image quality and made comparative MRI analysis difficult, leading to a low level of confidence from the blinded radiologist for the evaluation procedure. In future experiments, the use of a rigid cast should be validated to ensure that patients are scanned in the same position during repeated measurements.

Wrist denervation is classically one of the first surgical option offered for severe and persistent wrist OA. It is the only surgical procedure that preserves the joint anatomy and mobility of the wrist. Although this surgical procedure has shown globally encouraging results in chronic wrist OA, results are heterogeneous regarding different parameters such as improvement in mean pain scores (36–92%), or increase in grip strength (7–64%) and failure rate (re-operation, persistent pain or worse,6 to 29%) [70]. In our study, intra-articular injection of MF-PRP resulted in a 56% improvement in pain VAS and a 32% improvement in grip strength. Interestingly, Erne et al. compared the efficiency of autologous fat injection with the standard surgical procedure (Lundsborg’s resection arthroplasty) in the treatment of carpometacarpal OA of the thumb [71]. Adipose tissue injection was associated with a shorter delay before obtaining a complete relief of pain. Moreover, injection of MF-PRP is less invasive and likely to have a curative effect, allowing a potential regeneration of the cartilage whereas wrist denervation is a purely palliative analgesic treatment.

The major limitation of our study is the absence of a placebo group, which is a significant weakness given the considerable placebo effect in OA care [72]. Indeed, studies comparing long-term results with more accessible and cost-effective injectable drugs (steroid, HA or PRP alone) will be necessary to ensure the value of MSC-PRP injection in wrist OA. Given the complexity and size of the wrist articular cartilage, MRI quantification is complex and requires technological improvements to provide more valuable and objective data of the structural effects of MF-PRP injection on wrist cartilage. Finally, some of the patients included presented both radiocarpal and midcarpal wrist OA, whereas the MF-PRP injection was systematically performed in the radiocarpal joint. Further investigations of possible communication between both joints are necessary to ensure proper product distribution, especially when patients present associated SLAC or SNAC. This is further strengthened by the absence of clinical significance between these two groups. This issue should be clarified before conducting future studies.

Moreover, no data is available to clearly explain the mechanism of action of MF-PRP in OA although the main hypothesis is linked to anti-inflammatory properties as well as regenerative capacity likely due to the ADSCs kept intact into MF and growth factors contained in the PRP.

## 5. Conclusions

MF-PRP injection in wrist OA is a safe procedure and provides significant clinical benefit on function and pain relief for up to 12 months. Larger and placebo-controlled studies are required to confirm whether these new strategies can improve pain relief and wrist function and induce significant and measurable structural benefit, ultimately delaying and avoiding the need for a major surgical procedure.

## Figures and Tables

**Figure 1 jcm-11-05786-f001:**
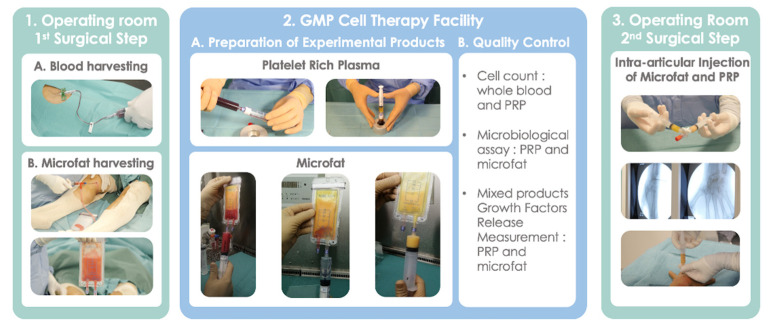
Procedure of preparation of experimental products: PRP and Microfat.

**Figure 2 jcm-11-05786-f002:**
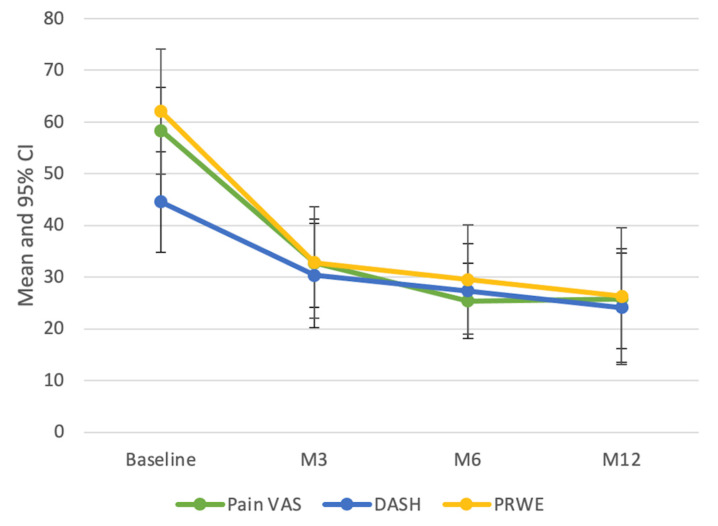
Evolution of DASH and PRWE scores and VAS of pain during the 12 months of treatment.

**Figure 3 jcm-11-05786-f003:**
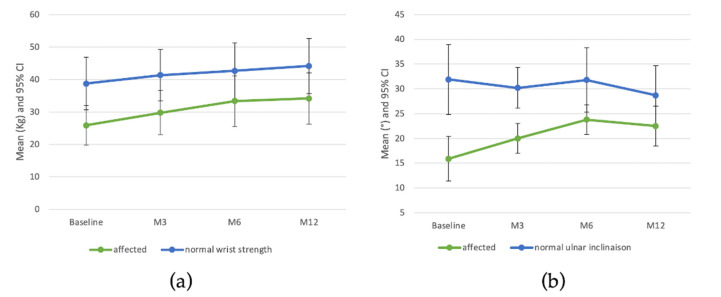
Wrist strength (**a**) and Ulnar inclination (**b**) over 12 months post treatment.

**Figure 4 jcm-11-05786-f004:**
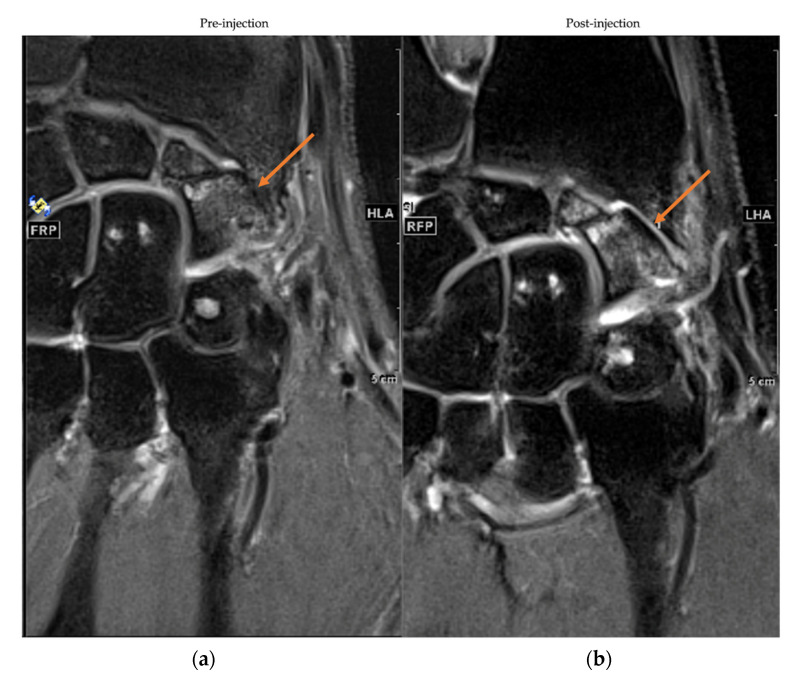
MRIs frontal sections of pre-injection (**a**) and 12 months post-injection (**b**). Cartilage condition was assessed with a 2D proton density turbo spin echo (TSE) with fat saturation sequence. The arrow indicates the location of the joint space of interest.

**Table 1 jcm-11-05786-t001:** Inclusion and Exclusion Criteria.

Inclusion Criteria
- Age between 18 and 75 years;
- Radio-carpal osteoarthritis grade 3 or 4 according to Kellgren and Lawrence classification resulting from post-traumatic malunion of an articular distal radius fracture or SLAC or SNAC;
- Failure of well-managed pharmacological treatment (analgesics, anti-inflammatory drugs, splinting, physiotherapy) defined as persistent daily painful condition > 40 mm according to visual analog scale (VAS);
- Body Mass Index ≥ 20 kg/m^2^;
- Hb > 10 g/dL;
- Negative pregnancy test;
- Written informed consent.
**Exclusion Criteria**
- Contraindications to MRI scanning; - Thrombocytopenia < 150 G/L; thrombocytosis > 450 G/L; thrombopathy; - Other coagulation disorders; - Infectious disease or positive serology to VIH-1, HCV, HBV and syphilis; - Chronic treatment with oral corticosteroids (or last dose taken less than 2 weeks before); - Intra articular wrist injection of corticosteroid or hyaluronic acid less than 8 weeks before inclusion; - Nonsteroidal anti-inflammatory drug or platelet inhibiting agent or antivitamin K treatment completed less than 2 weeks before injection; - Fever or recent disease (<1 month before injection); - Auto immune disease; - Inflammatory arthritis; - Immune deficit; - History of or ongoing malignancy; - Pregnancy; - Patient under guardianship or involved in another clinical trial.

MRI: Magnetic Resonance Imaging; Hb: haemoglobin; VIH: human immunodeficiency virus; HCV: Hepatitis C Virus; HBV: Hepatitis B Virus.

**Table 2 jcm-11-05786-t002:** Baseline Characteristics of patients.

Number of Patients (n)	12
Gender (women/men)	4 (33)/8 (67)
Age (years)	53.8 ± 14.8 (24–68)
BMI (kg/m^2^)	25.6 ± 3.8 (20–30.5)
Dominant hand: Right-handed	12 (100)
Affected hand: Right/Left	7 (58)/5 (42)
Disease duration from diagnosis (years)	6.6 ± 9.1 (1–25)
Traumatic etiology:	12 (100)
- SLAC	4 (33.3)
- SNAC	4 (33.3)
- Radius malunion	4 (33.3)
Osteoarthritis grade (Kellgren and Lawrence classification), (0 to 4)	Grade 3: 3 (25) Grade 4: 9 (75)
Localisation of osteoarthritis (X-ray)	
- Radiocarpal joint	8 (67)
- Radiocarpal joint and midcarpal joint	4 (33)

Data are presented as mean ± standard deviation and (minimum–maximum) for continuous variable and n (%) of patients for categorical variables. SLAC: scapholunate advanced collapse; SNAC: Scaphoid nonunion advanced collapse.

**Table 3 jcm-11-05786-t003:** Biological characteristics of experimental products.

	Mean ± SD	Median (Min–Max)
Total mean volume (mL)	3.6 ± 0.4	3.8 (2.7–4)
**Microbiological Assay, Free of germ**		
Microfat	10 (83)	
PRP	12 (100)	
**Cells (millions; %)**		
Platelets	741 ± 162; 95.3 ± 2.8	715 (480–1094)
Red Blood Cells	33.3 ± 17.7; 4.5 ± 2.7	30 (17.5–76)
Leukocytes	1.2 ± 1.3; 0.1 ± 0.1	0.9 (0.1–4.1)
Ratio platelets/cc of fat	410 ± 88	407 (274–547)
**Inflammatory cytokines (pg/mL)**		
IL-1b	7.7 ± 16.6	1.8 (0–58)
TNFa	25.3 ± 23.5	18.3 (0.6–84.7)
IL-6	4669 ± 3187	3861 (344–9960)
IFN γ	4 ± 6	0.1 (0–18.9)
**Anti-Inflammatory cytokines (pg/mL)**		
IL-10	12.1 ± 8.5	11.1 (0–25.1)
IL-1ra	155 ± 149	125 (28–539)
Ratio IL1Ra/Il1b	88.7 ± 108.4	39.9 (1.6–378.6)
**Growth factors (pg/mL)**		
PDGF	519 ± 340	490 (62–1095)
VEGF-a	28.2 ± 31.3	24.7 (0–101.8)
EGF	131.8 ± 67.3	135.3 (5.3–214.4)
FGF-2	6828 ± 3345	5904.9 (2410–12,999)
TGFB1	5019 ± 2891	5095.2 (1092–9271)

Data are presented as mean ± standard deviation and (minimum–maximum) for continuous variable and n (%) of patients for categorical variables. PRP: Platelet Rich Plasma; IL: interleukin; TNF-α: tumor necrosis factor-α; IFN-γ: interferon-γ; IL1Ra: interleukin-1 receptor antagonist; EGF: epidermal growth factor; VEGF: vascular endothelial growth factor; PDGF: platelet-derived growth factor; FGF2: fibroblast growth factor 2; TGF-β1: transforming growth factor β1.

## Data Availability

Main data presented in this study are contained within this article and Appendix A. The data that support the findings of the study are available by a request to the corresponding author.

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
