# Peer review of "Intra Articular Injection of Autologous Microfat and Platelets-Rich Plasma in the Treatment of Wrist Osteoarthritis: A Pilot Study"

_jcm, 2022, doi:10.3390/jcm11195786_

Round 1

Reviewer 1 Report

This research focused on microfat (MF) and PRP injection treatment in a moderate and mild wrist osteoarthritis. It is a prospective study with 1 year follow-up.

I found some doubts and remarks I would like to point out:

-      -    In the introduction authors comment on several operative treatment techniques (like in line 54 – wrist denervation) which are incomparable in the term of advancement of the disease with the treatment studied in the research. It should be better balanced and commented that several of these procedures are used in more advanced stages, when the non-operative treatment failed

-     -     In the procedure of injection “dorsal side of the wrist” should be used instead of “upper side” and the approach to the radio-carpal joint could be more precisely described (radial, central, depending on the pathology pattern)

-    -      Moreover, 4 patients had midcarpal (not “mediocarpal”) localization of OA changes (table 2) – how authors comment radiocarpal injection in these cases and maybe comment on influence of that fact on the result. These patients should be considered to be not included into a study or injection site should be modified

-      -    In the evaluation tools DASH is used where the hand and wrist is not sufficiently represented

-      -    In the same section (line 211 - 212) authors should described what kind of dynamometry they used to measure wrist strength because these methods are not commonly used and usually need dedicated equipment

-     -     authors conclude that this type of the therapy can be alternative to surgical procedures which neither is not supported by the study itself nor comparable or applicable in real life where surgical methods have well described and longer lasting (then 1 year follow-up) results and are used in more advanced stages of OA

The study is interesting and can add new information on the methods of treatment in mild and moderate wrist arthritis.

Author Response

Dear Reviewer 1,

We thank you for all your insightful comments
You will find attached the document answering point by point to your different remarks

Yours sincerely,

Jeremy Magalon, PharmD, PhD

Reviewer 2 Report

The study explores the safety of MF-PRP injection for wrist OA. This is a well-structured and clearly written manuscript. The subject matter should be of interest to clinicians and should help inform future controlled studies on wrist OA treatment. I have some minor suggestions:

1. Please clearly state in the abstract and study design section that this study is uncontrolled (no placebo group) and therefore cannot directly imply efficacy of the treatment.

2. Please recheck the inclusion criteria; BMI between 20 kg/m2 and [?]

Author Response

Dear Reviewer 2,

We thank you for your insightful comments
You will find attached the document answering point by point to your different remarks

Yours sincerely,

Jeremy Magalon, PharmD, PhD
